# Construction of a COVID-19 Pandemic Situation Knowledge Graph Considering Spatial Relationships: A Case Study of Guangzhou, China

Xiaorui Yang [1,2], Weihong Li [1,2,3,*], Yebin Chen [4] and Yunjian Guo [1,2]

1  School of Geography, South China Normal University, Guangzhou 510631, China
2  SCNU Qingyuan Institute of Science and Technology Innovation, Qingyuan 511500, China
3  Guangdong Shida Weizhi Information Technology Co., Ltd., Qingyuan 511500, China
4  Research Institute of Smart City, School of Architecture and Urban Planning, Shenzhen University, Shenzhen 518060, China
*  Correspondence: liweihong@m.scnu.edu.cn

**Abstract:** The outbreak of COVID-19 (coronavirus disease 2019) has generated a large amount of spatiotemporal data. Using a knowledge graph can help to analyze the transmission relationship between cases and locate the transmission path of the pandemic, but researchers have paid little attention to the spatial relationships between geographical entities related to the pandemic. Therefore, we propose a method for constructing a pandemic situation knowledge graph of COVID-19 that considers spatial relationships. First, we created an ontology design of the pandemic data in which spatial relationships are considered. We then constructed a non-spatial relationships extraction model based on BERT and a spatial relationships extraction model based on spatial analysis theory. Second, taking the pandemic and geographic data of Guangzhou as an example, we modeled a pandemic corpus. We extracted entities and relationships based on this model, and we constructed a pandemic situation knowledge graph that considers spatial relationships. Finally, we verified the feasibility of using this method as a visualization exploratory tool in the analysis of spatial characteristics, pandemic development situation, case sources, and case relationships analysis of pandemic-related areas.

**Keywords:** COVID-19; spatial relationship; knowledge graph visualization; relationship extraction



## 1. Introduction

COVID-19 is characterized by strong contagiousness, rapid mutation, and numerous mutation types, all of which have contributed to its serious impact on human health [1–3] and the social economy [4,5]. According to World Health Organization (WHO) real-time statistics (Appendix A), as of 29 July 2022, 571,198,904 cases of COVID-19 have been confirmed globally, and 6,387,863 people have died. The infectious diseases can be effectively controlled by studying the law of infectious disease transmission [6].

Statistical models and geographical methods have been widely used in the analysis of infectious disease dissemination. In 1927, [7] put forward a complete mathematical model for studying the spread of infectious diseases and established the first susceptible-infected-recovered (SIR) warehouse model of infectious diseases. Based on SIR, [8] have conducted further research and increased the categories of exposed persons in terms of population division, which is named the susceptible-exposed-infectious-recovered (SEIR) model. Applying the SEIR models to COVID-19, [9] found the individuals had a long incubation period that took time to be exposed; that is, when the person was infected but not yet infectious. Through epidemiological investigation and analysis of cases initially reported in Wuhan, [10] found the spot that may be the early outbreak place of the pandemic. By calculating the distribution parameter, significant heterogeneity was found, which was mainly due to the existence of super communicators [11]. The ecological niche models

and spatiotemporal analysis were used to detect the significant space-time clusters of the pandemic [12,13]. Spatial autocorrelation and Spearman's rank correlation methods were used to prove that the global spatial autocorrelation was extremely significant for the cumulative cases at county levels in Hubei province [14,15].

A knowledge graph (KG) is a structured representation of facts, consisting of entities, relationships, and semantic descriptions. Its purpose is to describe entities, concepts, events, and their relationships in the physical world [16]. There are two types of KGs according to their scope of application: the general KGs [17], and the domain KGs [18]. General KGs include language KG, commonsense KG and encyclopedic KG, and a domain KG is based on domain-specific data, which contains more complex knowledge and structures [19].

With the outbreak of COVID-19 and the increasing use of KGs, the application of KGs to the study of COVID-19 has gradually attracted the attention of both domestic and overseas scholars. The COVID-19 KGs, which can be described as domain KG, are different from the mathematical transmission models [9,20,21] and the simulation transmission models [22–24]. The former can visualize and mine the relationship and characteristics of things related to the pandemic situation, while the latter is more used to predict the relationship between the number of cases and time. Specifically, the main purposes for which KGs are applied to the field of COVID-19 research include (1) knowledge discovery and interpretation through case relationships and case activities, and providing relevant decision support [25–29]; (2) knowledge question-answering and queries, or constructing a knowledge base similar to an academic discipline [30–32]; (3) analyzing public opinion and judgment [33–35]; (4) discovering effective therapeutic modalities and drugs [36–38]. In addition, pandemic situation KGs based on the activity data of COVID-19 cases can assist in the identification of transmission relationships, analysis of temporal and spatial patterns of the pandemic, simulation of pandemic transmission, and assessment of pandemic risk. Existing studies have been concerned with case activity events, have constructed the ontology and data layers of the case KG according to the constituent elements of the population activity model, and have used natural language processing (NLP) toolkits to extract the basic situation of the case, case activities, and case semantic relationships [25,26]. Some scholars have also used deep-learning algorithms to extract case relationships [27] and entities [39] from textual matters about the pandemic. Some researchers have constructed case relationship graphs based on case flows and their inter-relationships to reveal infection patterns and identify potentially infected persons [40,41]. Ref. [28] developed a pandemic spread graph for the United States and Japan, associating states (cities) with administrative divisions at the same or lower level. Ref. [25] added an inclusion relationship between geographical entities, but neglected the problems arising from a single geographical entity having multiple nodes. Refs. [29,42] used Twitter data to extract theme, date and event relationships, and analyze how the Twitter data reflected public emotions concerning COVID-19-related topics. The above researchers analyzed the pandemic situation, case traceability, virus diffusion path and public opinion information using KGs, which thus played a positive role in the prevention and control of COVID-19. However, these researchers seldom considered the spatial relationship of geographical entities, which cannot be ignored when using pandemic situation KGs to explore spatial characteristics.

Knowledge extraction (KE) is to extract knowledge from multiple data sources, which are the basis for building a KG. KE includes named entity recognition (NER) and relationship extraction (RE). In the 1990s, NER and RE were mainly realized by manually writing rules and templates [43–45]. Machine learning was used to extract entities in the 2000s [46]. A single-layer convolutional neural network (CNN) model was used for NER in 2011 [45]. In recent years, the application of the transformer [47] model to NER has also become the focus of research, including the Transformer Encoder for NER (TENER) model [48] and the Bidirectional Encoder Representations from Transformers (BERT) model [49]. Supervised [50,51], semi-supervised [52,53] and unsupervised [54–56] machine learning methods are also used in RE. The RE methods based on deep learning can be divided into the pipeline method and the joint extraction method. The pipeline method extracts information in two steps, first extracting entities, then extracting relationships, and finally

integrating triples for output [57,58]. The joint extraction method combines NER and RE models and extracts entity relationship triples from the text directly [59,60]. As one of the most popular and notable state-of-the-art language models, BERT is also used in entity relationship joint extraction. Ref. [61] utilized and fine-tuned the pre-trained BERT model for the construction of KGs without language dependencies, extracting new relationships with the BERT-based relation extraction model and integrating them into the KG. Ref. [62] constructed a BERT–BiLSTM-based entity relationship extraction model for food public opinion, in which the entity relationship types extracted by the BERT model were integrated into the BiLSTM and the corpus was described in Chinese.

In two-dimensional space, spatial relationships include distance relationship, direction relationship and topological relationship. Distance relationship describes the distance between geographical entities, including qualitative distance and quantitative distance. Quantitative distance is usually calculated by mathematical formulas, such as Euclid distance, Manhattan distance, and cosine distance. Qualitative distance is described by natural language [63]; in some knowledge-based geospatial analysis and mining systems, the number of the required qualitative distance words depends on the level of granularity, and quantitative distance implies different qualitative distances for different persons [64]. Direction relationship description methods can be divided into three categories [65–67]: cone model, projection-based model, and the Voronoi diagram-based model. Point-set topology [68], region connection calculus [69], symbolic projections [70], minimum bounding rectangle(MBR) [71] and n-intersection model [72] are used to describe topological relationships.

In this study, taking the city of Guangzhou as an example, we used the pipeline method to extract entities and relationships. The entities (including geographical entities) in the Chinese pandemic notification texts were extracted with the Language Technology Platform of the Harbin Institute of Technology (HIT LTP, Appendix A). For non-spatial relationships, we have made a corpus using partial notifications. Based on this corpus, we extracted the relationships in the remaining data by fine-tuning the BERT model. For spatial relationships, we calculated the distance and orientation relationships between the extracted geographical entities. We constructed a pandemic situation KG for COVID-19 that considers spatial relationships with these entities and relationships, to remedy the inadequacy of the existing COVID-19 graphs concerning spatial characteristics analysis and visualization and therefore provide more effective support for the prevention and control of COVID-19 in different regions and at different times.

## 2. Materials and Methods

### 2.1. Study Area

Guangzhou is located in the south-central part of Guangdong Province, with a huge and increasing floating population and complex demographic composition, and it is known as the South Gate of China. According to GaWC2020 (Globalization and World Cities 2020, Appendix A), Guangzhou is one of the world's first-tier cities, with a developed economy, transportation and information network; it also has superior medical conditions and relatively comprehensive case collection. During the pandemic, satisfactory achievements were obtained due to effective preventive measures in Guangzhou. For these reasons, Guangzhou is representative and we choose it as the study area. The date covers the period from 21 January 2020 to 28 May 2022. The study area and distribution of the places where the domestically transmitted cases were confirmed during the period are shown in Figure 1.

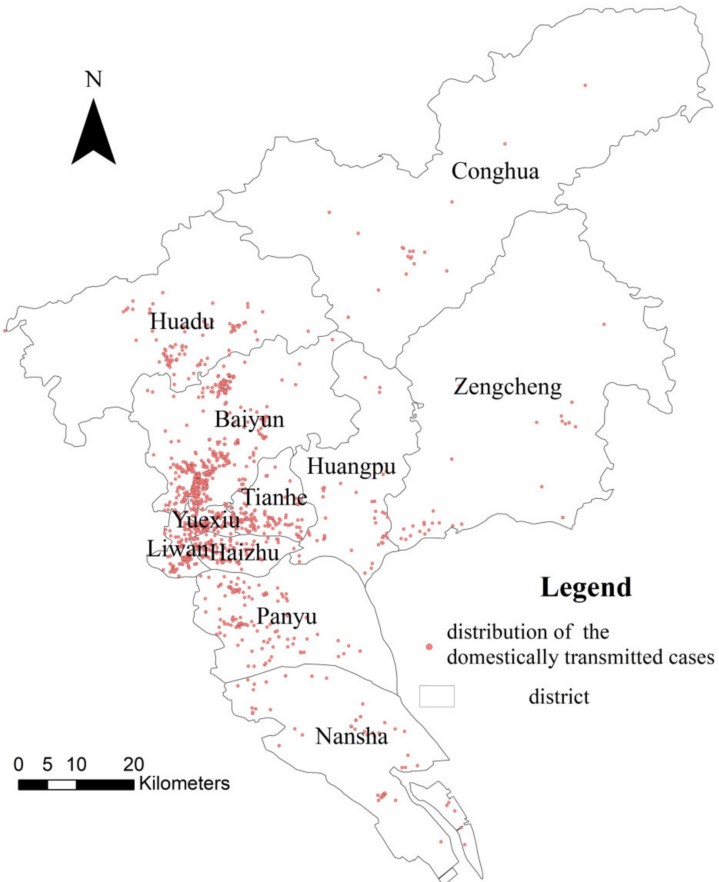

**Figure 1.** Administrative division map of Guangzhou. The map source is the standard map website (Appendix A). Where the red dots indicate the distribution of the places where the domestically transmitted cases were confirmed, each dot corresponds to one or more cases. The data was based on the results that we extracted from the notification texts that we mention below.

*2.2. Data Collection*

Table 1 shows the data used in this study, including pandemic data and geographical data. The Description column describes the specific data item, the Source column the source of the data item, and the Content column the data format or information type for each data item. The Count represents the data quantity. The raw data is described in Chinese. When a specific entity is mentioned below, we will use ID or English instead. Figure 2 shows the collection and processing of the data.

**Table 1.** Data source.

| Data Type | Description | Source | Content | Count |
|---|---|---|---|---|
| Pandemic | Guangzhou pandemic notification data | Guangzhou Municipal Health Commission | Text | 934 |
| Geographic | Names of Guangzhou POIs | Gaode map API | Names of POIs | 421,740 |
| | POIs involved in pandemic data | Gaode pick coordinate system | Names and coordinates of POIs | 1391 |
| | Administrative division data | National Bureau of Statistics | Administrative division | 2800 |

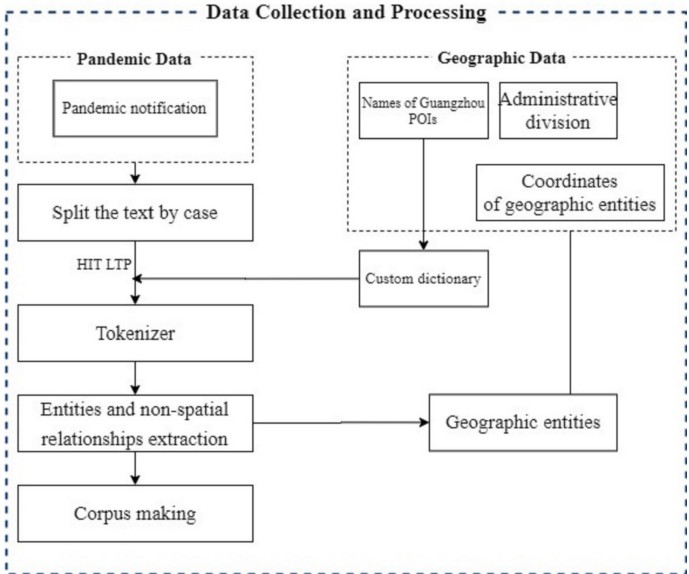

**Figure 2.** Data collection and processing.

2.2.1. Pandemic Data Collection and Processing

The pandemic data in this article were taken from the Guangzhou Municipal Health Commission website (Appendix A), and we used the bazhuayu collector software V8.5.2, Shenzhen, China (Appendix A) to obtain pandemic notification texts. Since the case data at the initial stage of the pandemic (before 21 January 2020) were not disclosed in Guangzhou, these cases were not considered in this study. The data processing includes: (1) sentence processing: we use the BERT model to extract the relationship between entities; the model, however, supports a maximum sequence length of 512. Each case data in the notification is usually separated into a paragraph, and python is used to extract each case data as a record and ensure that the length of each record does not exceed 512. At the same time, to make sure that the diagnosis date corresponds to the case correctly, the date information is added before each case record. We extracted 1487 domestically transmitted cases and 4932 imported cases from the 934 notifications; (2) corpus-making: after splitting the case record, HIT LTP combined with the custom dictionary is used to segment the text of each record. Based on the segmentation and further combined with manual recognition, we extracted several entity-relationship-entity triples from each text record. The records and their corresponding triples constituted the Guangzhou pandemic situation corpus (410 days of data for the time range from 28 February 2021 to 14 April 2022). Then the corpus could be used to train the model and improve the efficiency of non-spatial relationships and attributes extraction.

2.2.2. Geographic Data Collection and Processing

The geographic data acquisition and processing include (1) getting POIs names from the Gaode map API (Appendix A) and writing them into a .txt file after removing the duplicates to make a POI name dictionary, which helps us improve the accuracy of geographic entity recognition when using LTP; (2) querying and obtaining the pandemic-related geographic entities′ coordinates by the Gaode pick coordinate system (Appendix A), so that we can calculate the distance and azimuth relationship; (3) to identify the affiliation between administrative divisions at all levels, obtaining the administrative divisions at all levels in Guangzhou from the National Bureau of Statistics website (Appendix A).

*2.3. Methodology*

A KG can describe entities, concepts, events, and relationships between them in the physical world. Its basic unit is a triple composed of entity–relationship–entity. The

methods for NER and RE used in this study are described in detail in steps (3) and (4) below. The flowchart of constructing a COVID-19 pandemic situation KG considering spatial relationships is shown in Figure 3.

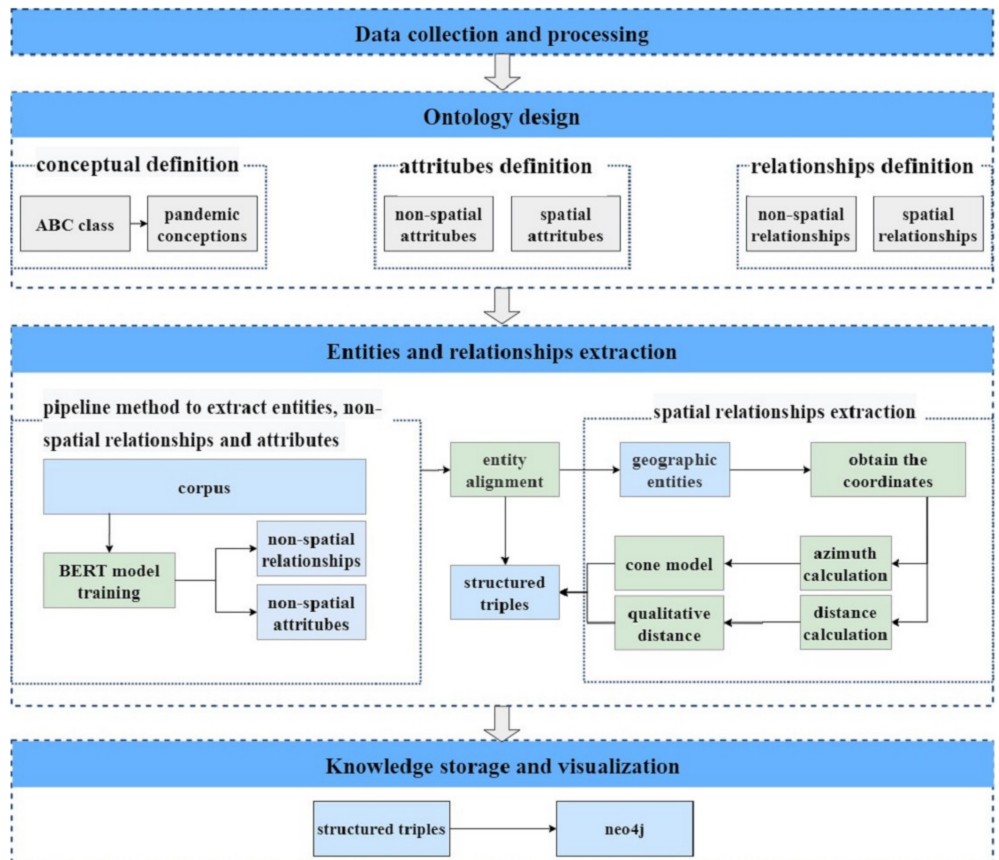

**Figure 3.** Flowchart of constructing a COVID-19 pandemic situation KG considering spatial relationships.

(1) Data collection and processing as described in Section 2.2.
(2) Ontology design of COVID-19 pandemic situation KG considering spatial relationships. In Section 2.3.2, we designed an ontology model, including designs of entity, attribute, and relationship.
(3) Entity, non-spatial relationship, and non-spatial attribute extraction in the notifications. We used the pipeline method to extract entity relationships. We extracted the relationship between entity pairs and entity pairs using two independent methods. In the entity pairs extraction stage, we used the LTP to manually identify the head and tail entities. In the relationship recognition stage, we fine-tuned the BERT model to extract the relationships contained in the pandemic notifications. Some of the same entities in the notifications use different words; we manually aligned these entities. Then we organized them into structured triples in tables, such as <h_entity (h_attr1, h_attr2 . . . ), relationship (r_attr1, r_attr2 . . . ), t_entity (t_attr1,t_attr2 . . . >, where the h_entity, t_entity and relationship refer to the head entity, tail entity and relationship between them; their respective attributes are in brackets.
(4) Spatial relationship extraction. Based on the geographical entities involved in the notifications, we further obtained the spatial coordinates of the geographical entities and then calculated the distance and azimuth between the entities. They were also organized into structured triples, such as <hg_entity (hg_attr1, hg_attr2 . . . ), s_relationship (sr_attr1, sr_attr2 . . . ), tg_entity (tg_attr1,tg_attr2 . . . >, where the hg_entity, tg_ entity and s_relationship refer to the head entity, tail entity and relationship between them in a geographic triple; their respective attributes are in brackets.

(5) Construction of pandemic situation KG considering spatial relationships. Based on the ontology design and structured triples, we coded the entities and assigned the same ID attributes to the same entities. Among these, the attribute of the spatial relationship stored the distance and direction information between the head entity and the tail entity. Finally, we used py2neo (a python library for neo4j) to store the structured triples and display them in neo4j.

### 2.3.1. Custom Dictionary

Using NLP methods to interpret unstructured texts involving geographic entities can discover and retrieve geographic information [73]. The NLP tool used in this study, the HIT LTP, adopts the pre-trained model and the multitask learning mechanism [74] and achieves only limited accuracy in identifying geographical entities. To improve the accuracy of geographical entity recognition, we defined the Guangzhou place name dictionary. We formed a dictionary from the administrative place names and POI names of Guangzhou, which we stored in a .txt file after duplicate entries were removed. In this mode, the model can preferentially perform named entity recognition, part-of-speech tagging, semantic role tagging, dependency syntax analysis, and semantic dependency analysis, according to the words in the dictionary.

### 2.3.2. Ontology Design

At the level of ontology design, we referred to the seven-step method [75] to construct the hierarchical and logical relationships of the ontology. We applied relevant concepts from the ABC ontology model [76], which is a general conceptual model that provides metadata descriptions of complex objects to facilitate interoperability between metadata ontologies in different domains.

The ontology concept design is shown in Figure 4. The original concepts of the ABC model are shown in gray fill, and our newly proposed domain-ontology concepts are white-filled; "place", "POI", "country", and "nation" are inherited from "Place", and represent the location or administrative division, the POI with coordinate information, and both the country and nationality of the source of overseas import cases, respectively; "transportation" is inherited from "Artifact" and represents the means of transportation; "infected" and "institution" are inherited from "Agent", and represent infected persons and relevant organizations, respectively; "profession" is inherited from "Work", and represents the occupation of the infected person; "pattern" is inherited from "Situation", indicating how the infected person is found; "event" and "isolation" are inherited from "Event" and represent the behavior event and isolation treatment mode, respectively, of the infected person.

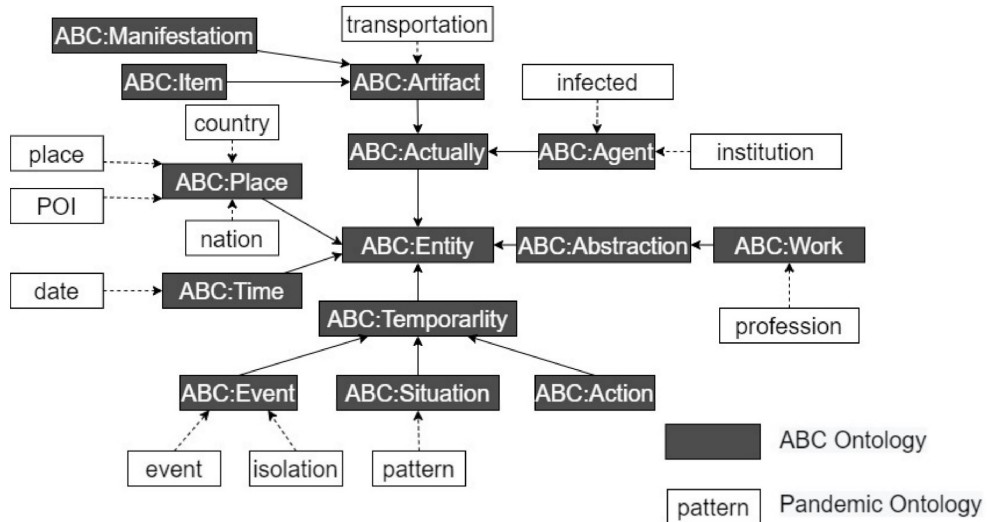

**Figure 4.** Ontology concept design.

Figure 5 shows the designed relationships between ontologies, and the meanings of various relationships are shown in Table 2. Among these, "infected persons" include asymptomatic infected persons and confirmed cases. According to the People's Government of Guangdong Province (Appendix A), controlled-level area means the community where the infected lives and the surrounding areas he/she often goes to, managed-level area means the place where the case worked and arrived from two days before diagnosis to isolation management, and prevented-level area means the area outside controlled-level and managed-level area in the country. To ensure that correct relationships could be established between ontologies and there is no ambiguity between entities, ID and name attributes were added to various ontologies, where "ID" is the unique identifier of each entity, and "name" is the name of the entity.

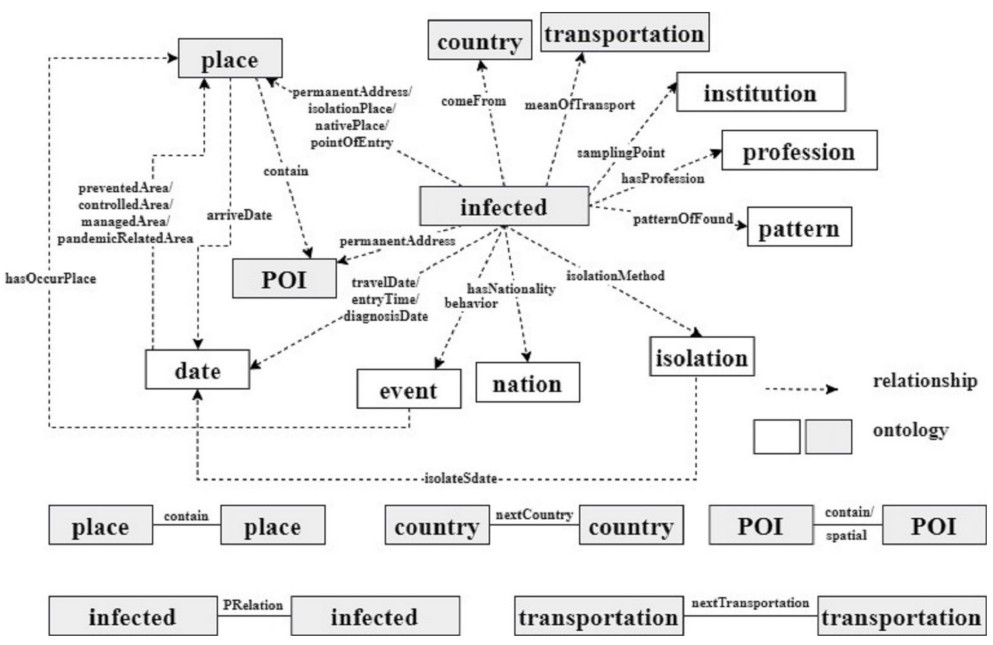

**Figure 5.** Ontology relationship design.

**Table 2.** Entity relationships and their meanings.

| Head_Entity | Relationship | Tail_Entity | Meaning |
|---|---|---|---|
| place/POI | contain | place/POI | wide range of administrative divisions includes administrative divisions at lower levels; administrative divisions or larger areas of POIs contain smaller areas of premises |
| infected | samplingPoint | institution | the facility that takes nucleic acid samples from infected individuals |
| infected | permanentAddress | place/POI | common residence of infected persons |
| infected | travelDate | date | date the infected person traveled |
| place | arriveDate | date | date of arrival at a place |
| event | hasOccurPlace | place | the place where an event or action takes place |
| infected | patternOfFound | pattern | how infected person is found to be infected |
| date | controlledArea | place | the date when a place is designated "controlled-level" |
| infected | isolationMethod | isolation | how infected are isolated |
| infected | isolationPlace | place | place of isolation for infected persons |
| date | managedArea | place | the date when a place is designated "managed-level" |
| date | preventedArea | place | the date when a place is designated "prevented-level" |

**Table 2.** *Cont.*

| Head_Entity | Relationship | Tail_Entity | Meaning |
|---|---|---|---|
| infected | hasNationality | nation | nationality of the infected person |
| infected | behavior | event | events experienced by the infected person |
| infected | nativePlace | place | the native place of the infected person |
| infected | meanOfTransport | transportation | mode of transportation of the infected person |
| isolation | isolateSdate | date | the date the infected person starts isolation |
| infected | comeFrom | country | country or region where the infected person comes from |
| infected | pointOfEntry | place | place of entry of the infected person |
| infected | entryTime | date | date of entry of the infected person |
| date | pandemic-RelatedArea | place | affected areas on a given day |
| transportation | nextTransportation | transportation | order in which the infected person traveled |
| country | nextCountry | country | order in which an infected person passed through a country or region |
| infected | diagnosisDate | date | when the infected person was diagnosed |
| infected | hasProfession | profession | infected person's occupation |
| infected | PRelation | infected | relationship between cases |
| POI | spatial | POI | the spatial relationship between POIs |

The data layer finally formed according to the above ontology relationship design is shown in Figure 6.

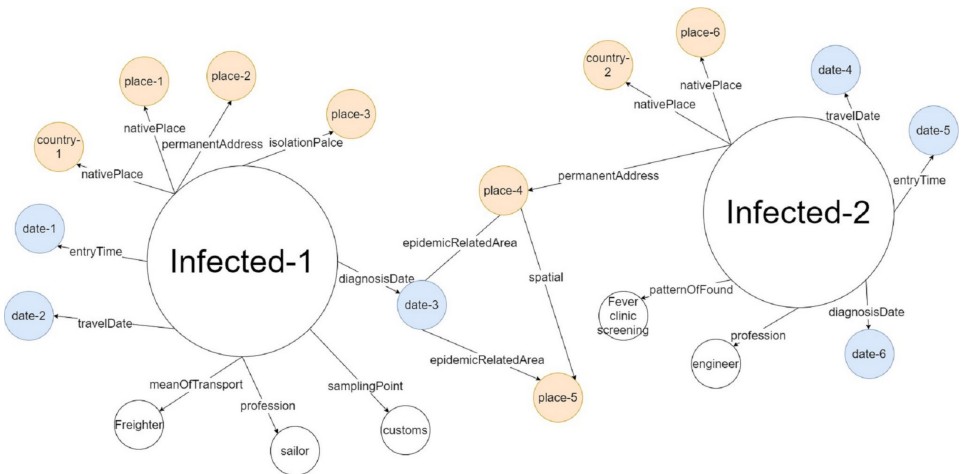

**Figure 6.** Data layer of the pandemic KG.

### 2.3.3. Spatial Relationship Design of Geographic Entities

The direction relationship and distance relationship are considered. We adopted the improved cone model [77] to determine the direction relationship. We got the corresponding direction relationship by calculating the azimuth with two-point coordinates, as shown in Figure 7a, and Figure 7b shows how it is stored in the graph.

We calculated the distance relationships in terms of both quantitative and qualitative distances, and their storage in the graph is shown in Figure 7b. The quantitative distance can be calculated by Euclidean or Haversine methods. If the coordinates were projected, the Euclidean distance was calculated as Formula (1): 1.

$$distance = \sqrt{(x_1 - x_2)^2 + (y_1 - y_2)^2} \tag{1}$$

where $(x_1, y_1)$ and $(x_2, y_2)$ represent the projected coordinates of the two POIs. If the coordinates were geodetic, the Haversine method shown as Formula (2) was applied:

$$distance = 2r \cdot arcsin\sqrt{sin^2\frac{(Lat1 - Lat2)}{2} + cos(Lat1) \cdot cos(Lat2) \cdot sin^2\frac{(Lng1 - Lng2)}{2}} \quad (2)$$

where $Lng1$ and $Lat1$ represent the longitude and latitude of point A, $Lng2$ and $Lat2$ represent the longitude and latitude of point B, and r is the radius of the earth.

We classified qualitative distance as near, medium, or far, as follows:

$$dist\_level = \begin{cases} near & distance \leq D_1 \\ medium & D_1 < distance \leq D_2 \\ far & D_2 < distance \leq D_3 \\ very\ far & D_3 < distance \leq D_4 \\ \dots & \dots \end{cases} \quad (3)$$

In this article, we used 1500 m as the maximum distance. If the maximum distance between two points was not more than 1500 m, the spatial relationship between those two points was explicitly established: we set the $D1$ value as 500, the $D2$ value as 1000, and the $D3$ value as 1500.

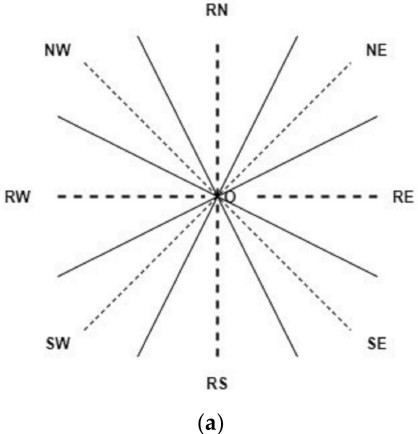

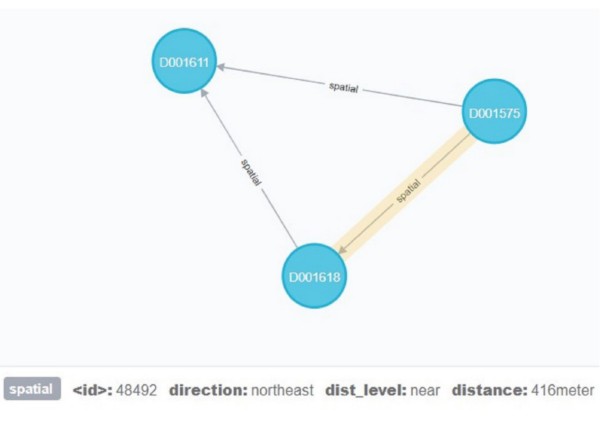

spatial  **<id>:** 48492  **direction:** northeast  **dist_level:** near  **distance:** 416meter

(a)    (b)

**Figure 7.** Improved cone model and the representation of spatial relationships in KG: (**a**) where direction = (RN, NE, RE, SE, RS, SW, RW, NW, O), and O is the same-position relationship; (**b**) the tail entity (arrow) acts as the origin, the "direction" attribute indicates the orientation of the head entity (arrow tail) relative to the tail entity (arrowhead), as shown in the figure, the entity with ID "D001575" is located in the northeast of the entity with ID "D001618", and the quantitative distance and qualitative distance between them are 416 m and near, respectively.

### 2.3.4. Non-Spatial Relationship Extraction Based on BERT Model

BERT is based on the bidirectional transformer and is used to represent word vectors. After fine-tuning, it can automatically learn the feature in sentences, obtain the expression of sentence vectors, and extract relationships from COVID-19 notification data. How the model works will be introduced in the following two aspects: input representation and relationship extraction.

(1)    Model input representation

An input example of the BERT model is shown in Figure 8. Given a sentence, in the token embedding, a word dictionary is built, and $[CLS]$ and $[SEP]$ are added at the beginning and end of the sentence, respectively, to represent the beginning and end of the sentence. The output of the last transformer layer that corresponds to $[CLS]$ plays the role of aggregating the features of the whole sentence, which means it is a semantic feature vector that can represent the sentence. In the segment embedding, the learned embedding

is added to each tag to calibrate the context relationship of the two clauses; in the position embedding, the vector representation of each position is learned to include the sequence characteristics of the input sequence. Finally, the three embeddings are added to obtain the final BERT input representation.

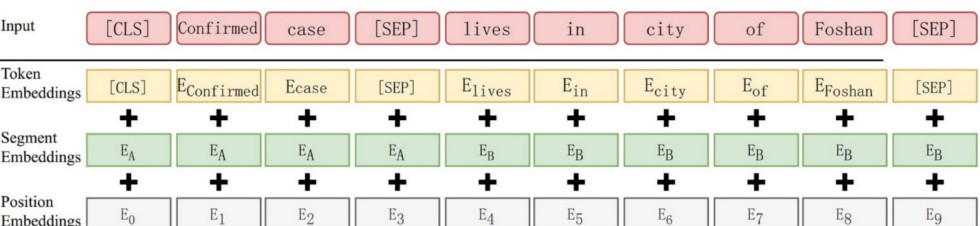

**Figure 8.** BERT input representation.

(2) Relationship extraction

Figure 9 shows the framework of the BERT relationship extraction model. First, we input the preprocessed sentences into the model. BERT then extracts the classification features, the sentences are encoded, and vector representations of sentences and individual words are obtained. After the BERT encoding, we select sentence vectors and entity vectors, obtaining their final vectors through the activation function and the full connection layer. The obtained vectors are then spliced, producing a relationship feature vector to be classified. Finally, we used this vector for SoftMax classification to obtain the relationship type R.

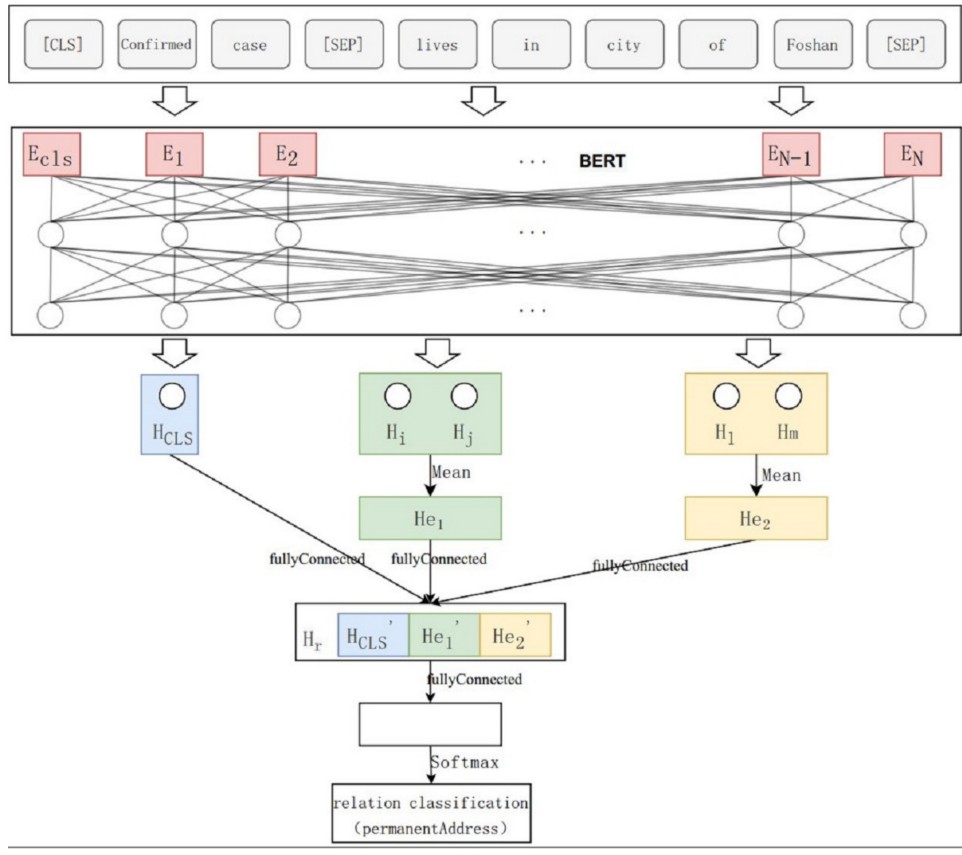

**Figure 9.** Relationship extraction based on BERT.

The final output after passing through the first fully connected layer is as follows:

$$H'_{cls} = W_0[\tanh(H_0)] + b_0 \tag{4}$$

$$H'_{e1} = W_1 \left[ \tanh \left( \frac{1}{j-i+1} \sum_{t=i}^{j} H_t \right) \right] + b_1 \tag{5}$$

$$H'_{e2} = W_2 \left[ \tanh \left( \frac{1}{l-m+1} \sum_{t=m}^{l} H_t \right) \right] + b_2 \tag{6}$$

where $H'_{cls}$, $H'_{e1}$, and $H'_{e2}$ represent the feature vectors obtained by $[CLS]$, the head entity, and the tail entity, respectively, through the full connection layer; *tanh* is the activation function; $W_0$, $W_1$, and $W_2$ represent the weight vectors of $[CLS]$, the head entity, and the tail entity, respectively; and $b_0$, $b_1$, and $b_2$ are the corresponding bias vectors; *i* and *j* represent the start and end positions of the head entity; *l* and *m* represent the start and end positions of the tail entity, respectively; and $H_t$ represents the state vector of a word in an entity. The $H_r$ vector is classified by the second-layer full connection layer and SoftMax to get the probability *P* of various relationships. The relationship with the largest probability value is taken as the relationship between the two entities.

$$p(y|x,\theta) = softmax(W * H_r + b) \tag{7}$$

Here, *y* is the total relationship type, $\theta$ is the parameter to be learned, $W \in R^{N \times 3d}$ *N* is the number of relationship types, and *b* is the bias vector.

The model was trained using the cross-entropy loss function *L*:

$$L = -\sum_{i=1}^{k} \log p(y_i|x_i,\theta) \tag{8}$$

where *k* is the size of the batch.

## 3. Experiments and Results

Based on 934 pieces of pandemic notifications, names of 421,740 POIs, 1391 pieces of POIs with coordinate attributes, and 2800 pieces of Guangzhou administrative division data, as shown above in Table 1, we conducted the following experiments in sequence.

### *3.1. Experiments*

3.1.1. Custom Dictionary Experiment

We used the custom dictionary recorded in Chinese when using LTP for word segmentation, and the comparison before and after is as follows:

Original sentence: " ... 居住在越秀区大塘街道德政北路雅荷塘社区 ... 即转广州医科大学附属市八医院隔离治疗。"

Word segmentation results before using a custom dictionary: [[ ... ... ' 居住', ' 在', ' 越秀区', ' 大塘街道德政北路雅荷塘', ' 社区', '。' ... ... ' 即', ' 转', ' 广州', ' 医科', ' 大学', '附属市八', ' 医院', ' 隔离', ' 治疗', '。']]

Word segmentation results after using a custom dictionary: [[ ... ... ' 居住', ' 在', ' 越秀区', ' 大塘街道', ' 德政北路', ' 雅荷塘社区', ... ... ' 即', ' 转', ' 广州医科大学附属市八医院', ' 隔离', ' 治疗', '。']]

After using the custom dictionary, the words " 越秀区- 大塘街道- 德政北路- 雅荷塘社区" and " 广州医科大学附属市八医院" were accurately identified on the administrative division level, which increased the accuracy of the automatic identification of geographical entities.

3.1.2. Non-Spatial Relationship Extraction Experiment

410 days of pandemic notification texts were made into a corpus, forming 3472 sentences, corresponding to 31,082 entity relationship pairs, covering 25 relationships other than "spatial" and "PRelation" in Table 2, as well as the two attributes of sex and age. PRelation describes the relationship between cases, and we got it by the keyword "close

contact". Using the created pandemic corpus as the training data of the model, the corpus was divided into training, validation, and test sets in a ratio of 6:2:2 (18600:6241:6241).

The pre-trained model we adopted was BERT-base Chinese (Appendix A), and the hyperparameters we used are shown in Table 3:

**Table 3.** Model hyperparameters.

| Parameter | Value |
| --- | --- |
| Epoch | 6 |
| Learning_rate | 0.002 |
| Batch_size | 4 |
| Dropout | 0.4 |

The final loss value of the model is 0.00025 (Figure 10a), with an accuracy level of 99.6% in the training set and 99.3% in the validation set (Figure 10b). When we tested the model on the test set data not used in the model training, the relationship recognition accuracy reached 95.0%. We then applied the trained model to the pandemic notifications for the remaining days to obtain the entity relationships.

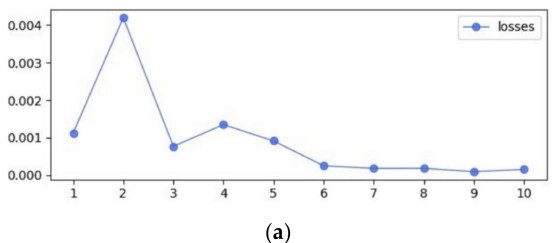
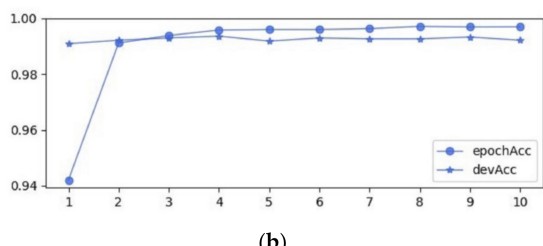

(**a**) (**b**)

**Figure 10.** Accuracy of the model: (**a**) the loss value on each epoch; (**b**) the accuracy of relationship recognition on each epoch training set and validation set. The loss value is almost unchanged after epoch = 6 when the relationship recognition accuracy is also at its highest.

### 3.1.3. Graph Construction

Based on the ontology and relationship design in Section 2.3, we used py2neo to store all entities and entity attributes in neo4j and then used the uniqueness of ID to establish the relationships between entities. The final pandemic situation KG of COVID-19 contained 12 types of entities and 27 types of relationships. The number of entities and relationships is shown in Figure 11.

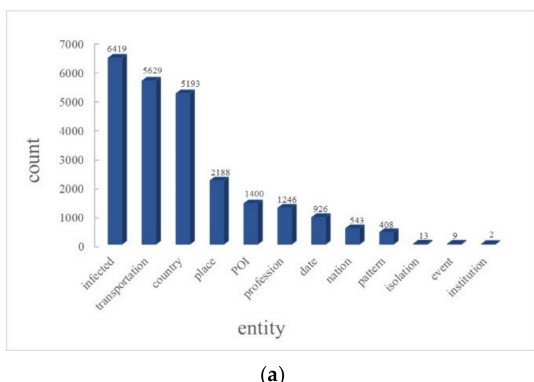
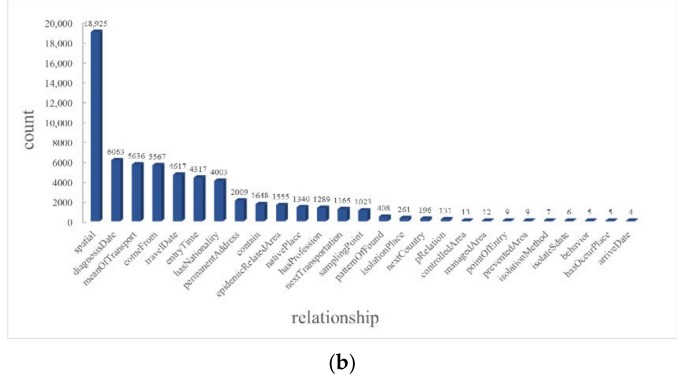

(**a**) (**b**)

**Figure 11.** The number of entities and relationships: (**a**) the number of entities of each type that we extract and store in the KG, the total number is 23976; (**b**) the number of relationships of each type that we extract and store in the KG. The total number is 60223.

### 3.2. Analysis of Experimental Results

3.2.1. Spatial Analysis of Pandemic-Related Areas

Taking domestically transmitted cases as an example, as shown in Figure 12a and b, the higher the population density, the more cases. We also conducted the spatial analysis with SaTScan (Appendix A) based on population, coordinate and case data, identifying the six cluster regions as shown in Figure 12c and Table 4. The type of analysis in the software is "Spatial", and the probability model is "poisson". The centroids of the clusters locate in Yuexiu, Tianhe, Baiyun and Liwan, and cluster 1 with a radius of 19.2 km covers most areas of Haizhu.

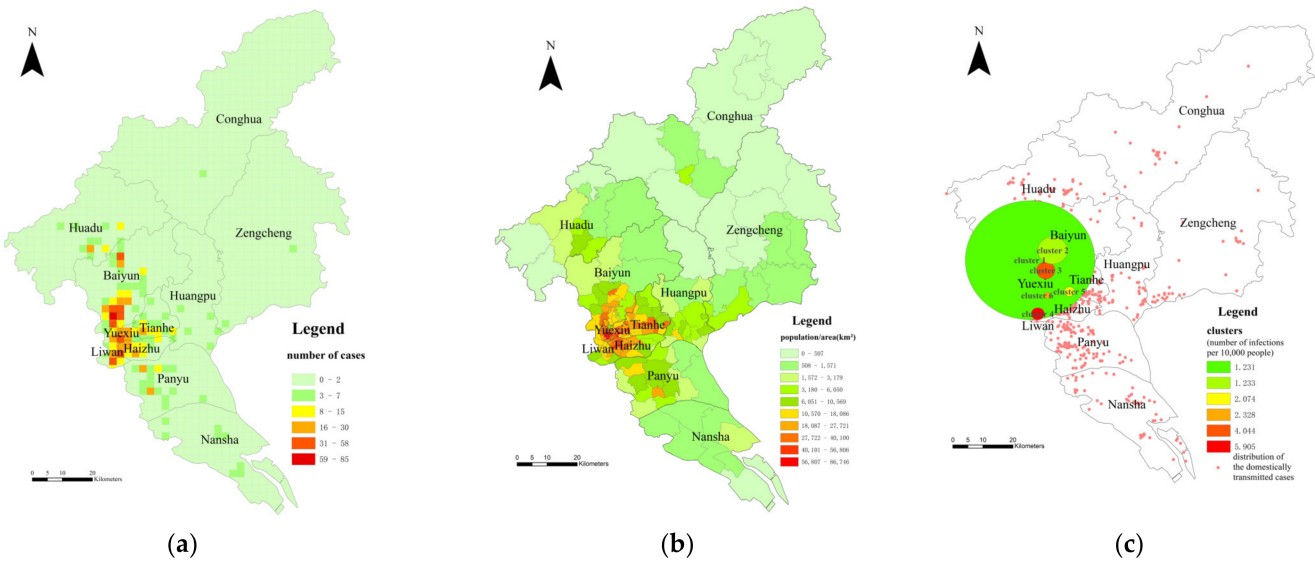

(**a**)  (**b**)  (**c**)

**Figure 12.** Distribution and clusters of the domestically transmitted cases and population density: (**a**) distribution of cases, in which the color indicates the number of cases, the data being based on notifications we collected and processed; (**b**) the population density of Guangzhou;(**c**) spatial analysis in SaTScan, where the circles represent the results of clustering.

**Table 4.** Spatial analysis in SaTScan.

| Cluster | Radius | Population | Number of Cases | *p*-Value | Location of Centroid |
|---------|--------|-----------|-----------------|-----------|---------------------|
| cluster 1 | 19.82 km | 8649558 | 1065 | $1.00 \times 10^{-17}$ | Baiyun |
| cluster 2 | 4.56 km | 875504 | 108 | $2.40 \times 10^{-2}$ | Baiyun |
| cluster 3 | 2.85 km | 484644 | 196 | $1.00 \times 10^{-17}$ | Liwan |
| cluster 4 | 2.01 km | 184581 | 109 | $1.00 \times 10^{-17}$ | Tianhe |
| cluster 5 | 1.56 km | 202505 | 42 | $1.70 \times 10^{-4}$ | Yuexiu |
| cluster 6 | 1.05 km | 210491 | 49 | $7.00 \times 10^{-7}$ | Haizhu |

In the KG, due to the establishment of spatial relationships between geographical entities, the spatial characteristics of pandemic distribution could be analyzed in terms of the spatial relationships between pandemic-related areas. As shown in Figure 13a, the nodes of pandemic-related areas that are close to each other are gathered together, and clustering characteristics in the pandemic areas of Yuexiu, Tianhe, Baiyun, Liwan and Haizhu can be found, which is consistent with Figure 12c. From a more microscopic perspective, as shown in Figure 13b–f, the spatial relationship between pandemic-related areas with clustering can be further presented in a small range. Furthermore, in Figure 13c, we noticed that Pazhou Street in Haizhu was a pandemic-related area for many days in February 2020; that is, cases were confirmed in the area for many days. The area again became a pandemic-related area in March 2022, which suggests that this might be an area especially prone to outbreak, meaning that more effective pandemic prevention policies

should be formulated according to the particular characteristics of the region and the flow of people therein.

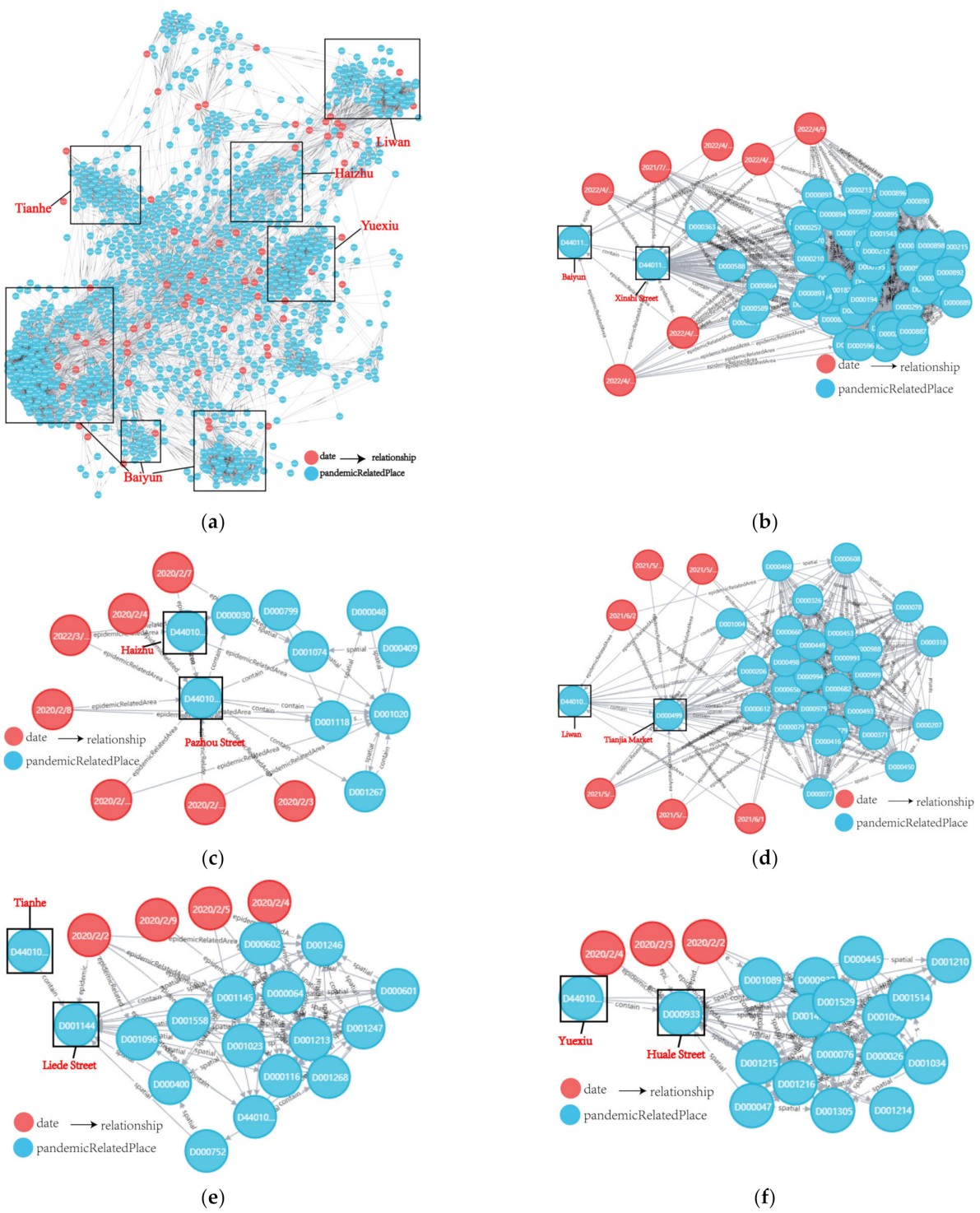

**Figure 13.** Spatial relationships between the pandemic-related areas: (**a**) spatial relationships between the pandemic-related areas in Guangzhou on all dates; (**b**) Xinshi Street in Baiyun; (**c**) Pazhou Street in Haizhu; (**d**) Tianjia Market in Liwan; (**e**) Liede Street in Tianhe; (**f**) Huale Street in Yuexiu.

### 3.2.2. Development of the Pandemic Situation

The pandemic in terms of daily numbers of infected persons can be viewed according to the relationship between cases and diagnosis time, as shown in Figure 14a. Here, we first identified the daily numbers of infected persons for the first week of April 2021. Second, we further clicked the case node to view the source, travel, and other information associated with these daily cases. By such means, we can analyze the cause of a particular outbreak, which can enable a quicker, more effective emergency response.

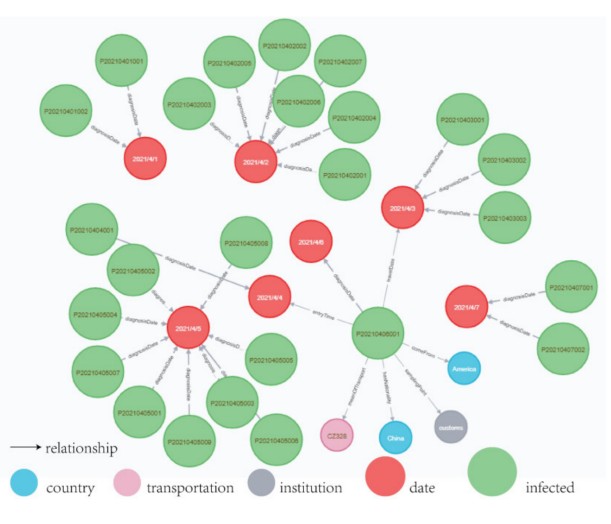

(**a**)

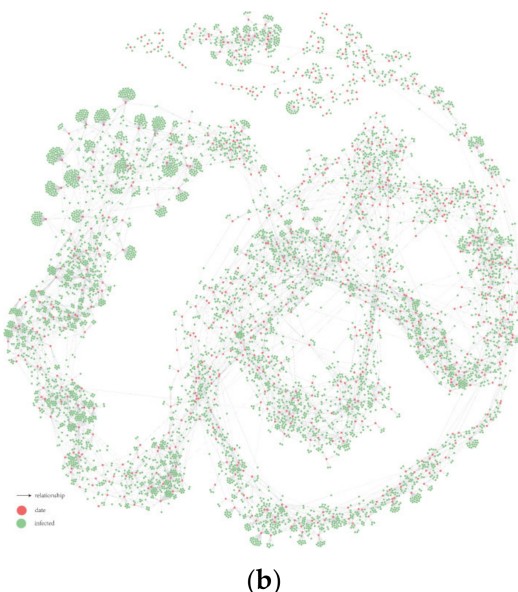

(**b**)

**Figure 14.** Development of the pandemic situation: (**a**) case situation in the first week of April 2021; (**b**) case profiles corresponding to all dates.

### 3.2.3. Sources of Imported Cases

Based on the constructed graph, we analyzed the source of infected persons. To illustrate, we produced a source map of cases in Guangzhou imported from abroad, as shown in Figure 15b. The top 30 countries where the imported cases came from are shown in Figure 15c. From the map and table, we observed that the United States, Canada, Japan, Russia, and other countries were the main source countries of imported cases from abroad in the period considered. Similarly, from Figure 15a, which shows the relationship between infected persons and their country of origin, we also analyzed the number of relationship lines connected to each country to reach the same conclusions as above. The graph shows the direct relationship between countries or regions and cases, which is conducive to further analysis of the situation involving infected persons.

Taking the United States, Britain, Bangladesh and Saudi Arabia as examples in Figure 16, when we further viewed the relationships between the infected and profession, we found that most of the infected persons from developed countries were overseas students, and most of the infected persons from developing countries were workers.

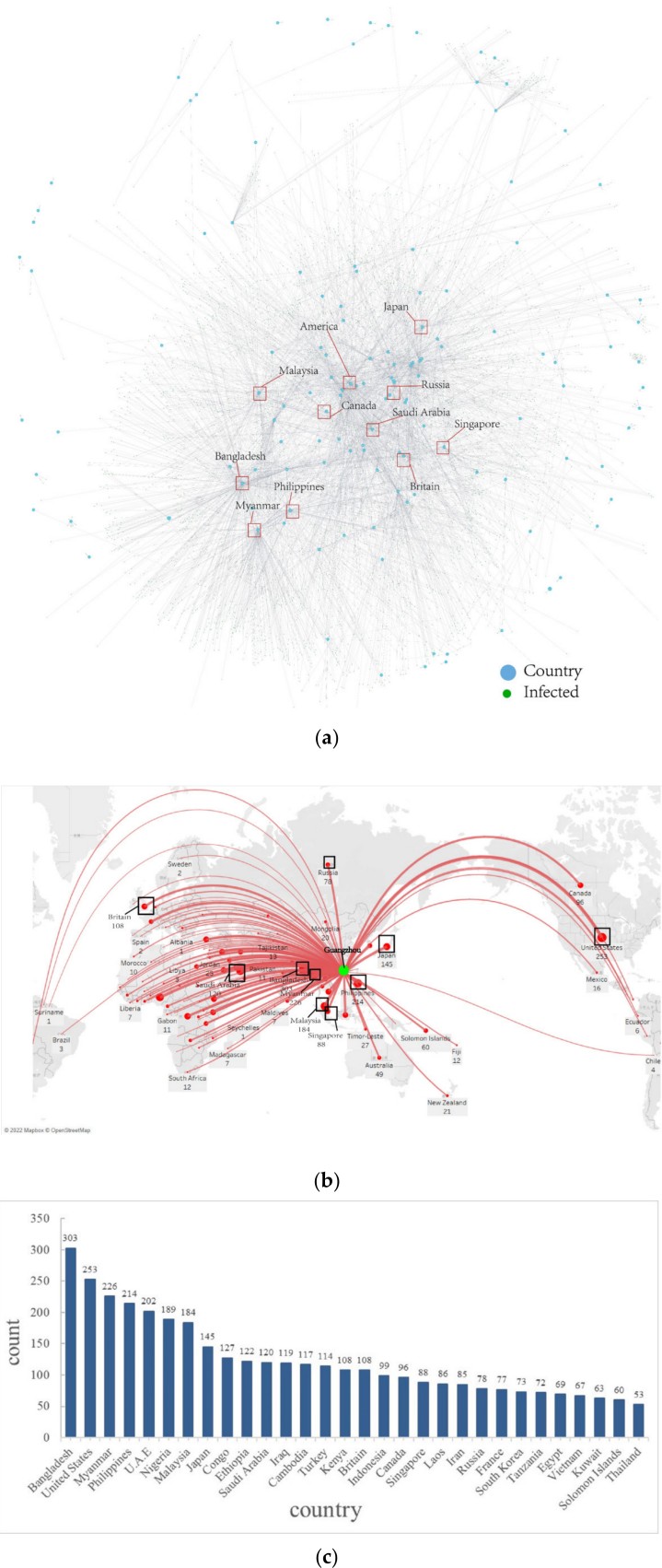

**Figure 15.** Sources of imported cases: (**a**) visualization analysis of case sources based on the KG; (**b**) case sources analysis based on the map; (**c**) top 30 countries where the imported cases came from.

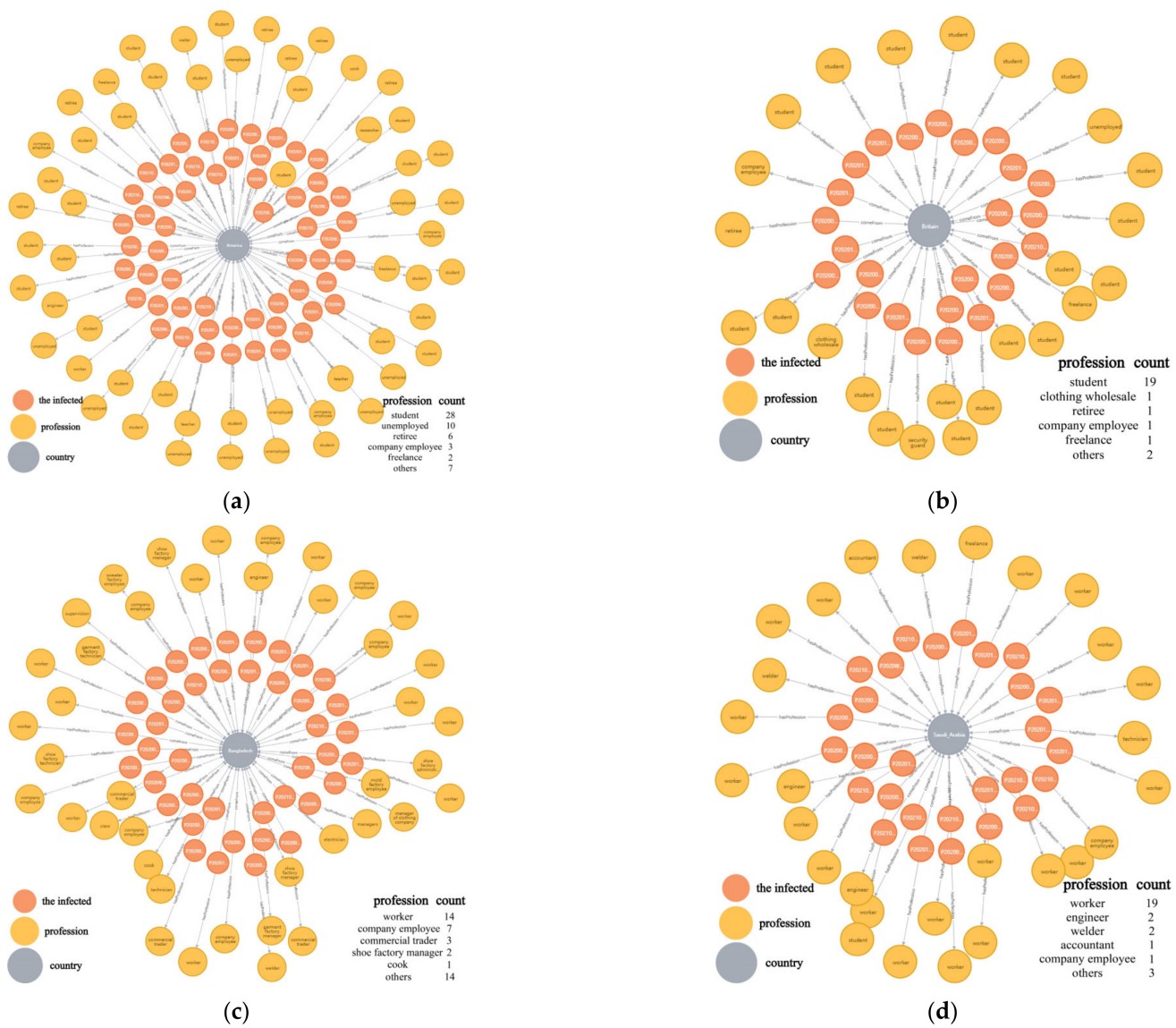

**Figure 16.** The relationships between the infected and profession, where the infected come from (**a**) America, (**b**) Britain, (**c**) Bangladesh, (**d**) Saudi Arabia. The figures only show the cases with occupational information; other cases without occupational information are not displayed.

### 3.2.4. Analysis of Case Relationships

By considering the relationships between the infected persons shown on the graph, we can analyze the infection paths of the pandemic and locate the source of infection. As an example, Figure 17a shows how case p20210528010 was the source of infection in this particular round. As shown in Figure 17b, we found that most of the infectious chains do not exceed level 4, which could be attributed to the rapid response of relevant departments. However, the information of some cases was not disclosed and the contact relationship between some cases was not provided explicitly. Although the characteristics of the infection chains could be analyzed through this method, this conclusion might not apply to any stage of the pandemic in Guangzhou.

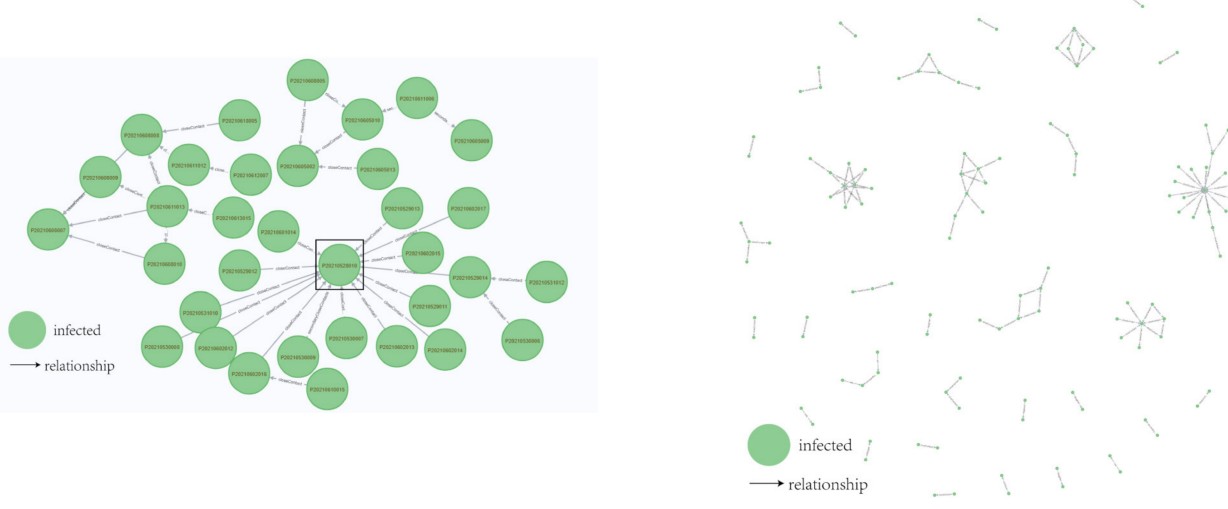

**Figure 17.** Case relationship visualization: (**a**) relationships between partial cases; (**b**) relationships between all cases.

## 4. Discussion

Comparing the graphs constructed in this study with those constructed by other researchers, we found that time is a necessary element in all methods, since time is an important factor in distinguishing and connecting different events; all events occur at a certain time, and the evolution path of events can be analyzed based on time [33]. In the mathematical model, time is an essential factor for analyzing and predicting the trend of the pandemic, and it is also an important factor for time-space analysis [9,13].

However, whether it was the case activity KG built with events [25,26], or a graph constructed using regional pandemic statistics data [28], we found more emphasis placed on the information mining and analysis of the cases. The graph constructed with the user data of social media software focused on mining public opinion information such as user emotions and policy changes [29] and gives less consideration to the spatial relationship between geographical entities, resulting in weak spatial feature analysis capability. The pandemic graph that considers spatial relationships and is described in this paper can identify more spatial characteristics while exploring case-source, intercase relationship, and pandemic area. From the macro perspective, we can find which districts have clustering characteristics, and we can also find out which places are prone to outbreak from the micro perspective.

In addition, the entity relationship extraction method we adopted in this study is efficient and accurate. The method described in this paper firstly extracts entity pairs from the results of word segmentation and then determines the relationship classification using the fine-tuning pre-trained language model BERT, which can quantitatively evaluate the extracted relationships and has an accurate extraction effect.

Based on the pandemic situation KG, the pandemic information can be retrieved, and the relationship between entities related to the pandemic can also be well visualized, such as the relationship between cases and their activities, and the relationship between cases. The super-infectious persons and potential infected persons, as well as more potential laws and knowledge, can be mined to provide decision support for stakeholders.

Our study has some shortcomings. First, we considered only the distance and direction relationships between pandemic spots when establishing a spatial relationship. Second, we ignored some data that was not disclosed, which might lead to different analysis results. In the future, we will explore how to add topological relationships, mapping points, lines, or planes into the graphs to facilitate more comprehensive spatial analysis and visualization, and Manhattan distance will also be used to compare with Euclid distance. What we used

for storage and visualization is a native graph database, and we have not yet developed an optimized visualization program; we will explore the method to better visualize the relationships between entities. We have constructed a pandemic situation KG based on pandemic and geographic data. How to organize the pandemic knowledge explored by mathematical models or geographical methods in the graph will be the focus of the following research, which could be used for knowledge Q&A, as well as more favorable decision support for the decision maker.

## 5. Conclusions

The current KG of COVID-19 does not consider the spatial relationships among geographical entities, and because of the shortcomings of the RE method, we designed a new method of extracting non-spatial and spatial relationships by comprehensively using the BERT model and spatial analysis theory to build a pandemic situation KG of COVID-19. We drew three conclusions from our study:

(1) Most of the cases in Guangzhou revealed spatial clustering characteristics, and the cases are mainly distributed in Yuexiu, Tianhe, Baiyun, Liwan and Haizhu, which are located in or close to the downtown and have a high population density; most of the imported cases were students from developed countries such as the United States and Britain, as well as workers from developing countries such as Bangladesh and Saudi Arabia.

(2) According to the disclosed notification data, the spread of COVID-19 in the Guangzhou population generally has not exceeded four generations. Most of the infected persons were close contacts or sub-close contacts of the "number one case", indicating that rapid government response effectively prevented the further spread of the pandemic.

(3) Compared with entity relationship extraction methods such as trigger word matching extraction and wrapper extraction, the entity relationship extraction of pandemic data achieved by the fine-tuned BERT model can be used to quantitatively evaluate RE accuracy, with relationship recognition accuracy for the Guangzhou pandemic reaching a level of 95.0%, thus indicating that the model has potential feasibility in the application of pandemic data entity-relationship extraction.

**Author Contributions:** Conceptualization, Xiaorui Yang and Weihong Li; Data curation, Xiaorui Yang and Yunjian Guo; Formal analysis, Xiaorui Yang and Yebin Chen; Funding acquisition, Weihong Li; Methodology, Weihong Li; Project administration, Weihong Li; Supervision, Weihong Li; Validation, Xiaorui Yang; Visualization, Xiaorui Yang; Writing—original draft, Xiaorui Yang; Writing—review & editing, Weihong Li and Yebin Chen. All authors have read and agreed to the published version of the manuscript.

**Funding:** This research was funded by [Ministry of Science and Technology of the People's Republic of China], grant number [2020YFF0303604], and in part by [Guangzhou Association for Science & Technology], grant number [20200115-8].

**Institutional Review Board Statement:** Not applicable. The research does not include any ethical concern and hence it does not require ethical approval.

**Informed Consent Statement:** Not applicable. The research utilized the available data from open-source, and the data we use does not include specific individual information.

**Data Availability Statement:** The main code, corpus and graph data have been uploaded to GitHub, which can be downloaded on https://github.com/youngxrui/Bert-In-Relation-Extraction-main2 (accessed on 22 August 2022).

**Acknowledgments:** We acknowledge Open Projects Fund of MOE Key Laboratory of Laser Life Science, College of Biophotonics, South China Normal University for its support. We thank professor Jizhe Xia from the School of Architecture and Urban Planning, Shenzhen University, for polishing the manuscript. We thank Jiachen Yan, Zhenduo Dou, Cheng Huang and Shugong Liu from the College of Biophotonics, South China Normal University for their work on the corpus-making. The anonymous reviewers are thanked for their comments that helped us to improve the paper.

**Conflicts of Interest:** The authors declare no conflict of interest.

## Appendix A

World Health Organization real-time statistics on coronavirus disease pandemic: https://www.who.int/emergencies/diseases/novel-coronavirus-2019 (accessed on 29 July 2022).

Globalization and World Cities 2020: https://www.lboro.ac.uk/microsites/geography/gawc/world2020t.html (accessed on 10 August 2022).

The standard map website: http://bzdt.ch.mnr.gov.cn/ (accessed on 25 July 2022)

Guangzhou Municipal Health Commission: http://wjw.gz.gov.cn/ztzl/xxfyyqfk/yqtb/index.html (accessed on 29 May 2022).

Bazhuayu collector software V8.5.2, Shenzhen, China: https://www.bazhuayu.com/ (accessed on 14 April 2022).

Language Technology Platform of Harbin Institute of Technology (HIT-LTP): http://ltp.ai/docs/index.html (accessed on 25 March 2022).

Gaode map API: https://lbs.amap.com/api/ios-sdk/guide/map-data/poi/ (accessed on 10 April 2022).

Gaode pick coordinate system: https://lbs.amap.com/tools/picker (accessed on 28 June 2022).

National Bureau of Statistics: http://www.stats.gov.cn/tjsj/tjbz/tjyqhdmhcxhfdm/2021/ (accessed on 25 March 2022).

People's Government of Guangdong Province: http://www.gd.gov.cn/gdywdt/zwzt/yqfk/kpxc/content/post_3764136.html (accessed on 14 April 2022).

BERT base Chinese: https://huggingface.co/bert-base-chinese/tree/main (accessed on 27 March 2022).

SaTScan software: https://www.satscan.org/ (accessed on 14 October 2022).

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
