# Peer review of "Construction of a COVID-19 Pandemic Situation Knowledge Graph Considering Spatial Relationships: A Case Study of Guangzhou, China"

_ijgi, doi:10.3390/ijgi11110561_

Round 1

Reviewer 1 Report

The paper presents an interesting topic using a knowledge graph to explore the spatial and temporal relationships of COVID-19 cases from pandemic notification data in the Guangzhou district area.
The references are adequate and current, except for the linked URL (as discussed later in this review in the “formatting” enumeration. However, the first paragraph of the introduction changes very abruptly to the actual explanation, and does not provide enough background context on the interest of the presented research. This aspect should be improved further.
The manuscript can be broken into two main parts roughly corresponding to section 2 (materials and methods) where the data collection and mining process is discussed, and section 3 (experiments and results) with the analysis of the results is presented.
The main criticism is that the manuscript focuses more on the data preparation than on the analysis of the results (the same issue identified in a reference in lines 377-378), and in the opinion of the reviewer this aspect should be corrected either reformulating the focus primarily on the data collection and processing, or developing the analysis further.
There are some methodological aspects in the data processing that should be addressed or clarified, presented below as an enumeration:
•    The case of the city of Guangzhou might be limited, and the choice of geographical scope should be justified
•    Pandemic data seems to consists in only 934 reported cases, and some justification should be provided whether they are enough or some (e.g. notification) bias is present
•    Section 2.2.1 should be developed further to explain the processes in more detail
•    It is not explained if language detection was used (it is assumed the language is Chinese as in the example in p.11)
•    It seems that training was performed on a single day (410) as can be deduced in line 288 and line 300?
•    The azimuth is relative to the origin and destination points, as unlike the distance is different (directional) from A to B than from B to A. Was the ontology able to support this particularity, for example storing the encoding in matrix form?
•    It would be necessary to correct by spatial covariates (e.g. population density)
•    Clustering is mentioned in line 412-413 but not quantified
Other methodological aspects less important that should be addressed:
•    In table 2, “managed-level” and “prevented-level” seems country-specific and should be described
•    Is not necessary to include the formula for Euclidian distance (p.8), it is assumed that the coordinates were projected or the distance was geodetic (in which case the formula does not apply)
•    The classification of distances (d1, d2, d3, d4) is introduced in p.9 but the intervals are not discussed until p.12
•    The case relationships in 3.2.4 describe that the infection chains do not exceed level 4, but it should mention that there can be missing data
The methodology used seems to perform correctly, in particular the named entity recognition and part of speech identification, but it would be interesting to compare the performance to other established methods such as geocoding or using a gazetteer instead of a dictionary.
After the discussion of the methodology, the results presented are more a visualization exploratory tool that a proper spatial analysis, and in the opinion of the reviewer should be either developed further or presented just as cases of study for stakeholders as a decision support system (as for example in figures 12/13/14/15 working as an user interface), in which case the extension of the section is excessive in contrast to the methodology discussion, or instead of focusing on using classical spatial statistics on the extracted spatial relationships instead of presenting multiple visualizations of data without spatial context.
This lack of focus in the application of the results of the knowledge graph, specially the spatial component, also translates to the discussion and conclusion sections, that fail to address the spatial information that is the scope of the journal.

Typos (not exhaustive)
•    Objective world (e.g. line 36 of p. 1, line 132…) might refer to the real/physical world?
•    “At home” in line 43 might confuse the reader
•    “data” instead of “date” in Figure 5 diagram
•    “countrty” in table 2 (5th line from the bottom)
•    Hypen in “pandemic-cRelatedArea” in table 2
•    “pPelation” in line 290 should be “pRelation”
•    “pandemics” instead of “outbreak” in line 330
Formatting
•    Linked URLs should be included in the bibliography or included as footnotes, instead of inline, as they make reading the text more difficult
•    Intermediate lines between the three sources in the geographic entry in table 1
•    Bars in figure 11 should be sorted
•    Data source of Figure 1 is missing (it is not clear if it is the result of the research discussed later in the text, or some official source)
•    Figures for cases from each country (lines 349-350) and figure 12 should be included or provide a bar chart

Reviewer 2 Report

This article may be suitable for consideration in a different journal. There is a minimal amount of spatial relationship described in the paper. In addition, the author can better explain some of the procedures. For example, Figure 12-14 are barely legible, chart in Figure 11 is not explained. The paper organization is can be improved. For example, 2.2.1&2 did not explain the "processing" as specified in the title. Overall, It is very difficult to follow and understand.

Reviewer 3 Report

This manuscript proposes a method for constructing a covid-19 knowledge graph with spatial relationships. The authors design an ontology for the pandemic, extract entities and their relationships using LTP and BERT models, and then extract the spatial relationships. A case study of Guangzhou is used for the demonstration of the proposed method.

The paper is well written, and the structure is clear. The methodology proposed is reasonable and the experiment shows useful patterns. Nevertheless, the review has the following comments:

1.       Since this study uses unstructured COVID-data as the test data, natural language processing is an important step. The authors only shortly discuss and compare the BERT model used in their study with other models in the section of Discussion and Conclusion. However, there are no prior information in the previous sections, e.g., the state of the art in Introduction. The author should add some information on the motivation why they choose such models like BERT but not the others.

2.       In Figure 12(a),13(a), and 15(b), the legends are not complete. Please replace them using complete figures.

3.       The numbering of 2 and 3 in Line 228 and Line 240 is quite confusing and are incorrect. Do they belong to section 2.3.4? What are they numbering after?

4.       What does the “the output of the last transformer layer” in Line 231 refer to in Figure 8? Please state it explicitly.

5.       What is the meaning of “pPelation” in Line 290?

6.       Are there any reasons to select 500m, 1000m and 1500m for distance categories, e.g., the size of the study region?  Furthermore, are the extracted spatial relations used in the experiment analysis?

7.       The id “p20210529010” in Line 368 seems to be inconsistent with the information in Figure 15 (a). Should it be “p20210528010”? Please correct the text or the figure to make it consistent.

8.       It would be great if the authors could point out some future work or research directions at the end.

Reviewer 4 Report

It is a good idea to construct the knowledge graph of COVID-19 considering spatial relationships. The paper put forward a method for constructing it and taking the COVID-19 and geographic data of Guangzhou as an example to verify it. There are some problems listed below:

1. Extensive editing of English language and style required.

2. Figure 3, the idea map should be optimized to better represent the logic of the method and now still has some mistakes in the figure, such as relation--raelation. From line 141--163, it is difficult to understand how the spatial relation extraction can be used in the knowledge graph construction? 

3. The names of the figures of this paper should be optimized.

4. The title of this paper, COVID-19 has different aspects(association), this title use COVID -19 is too broad.

Round 2

Reviewer 4 Report

The author revised the manuscript and now the paper is written well and addresses the planned hypothesis. The proposed method is appropriately validated through the experiment. The conclusions are borne out of the evaluation results.

Minor point: 1. Page 4 line 155, the data were...    the data was 

      2. The author should add a note that  the source of the map in figure 1.

Author Response

Response to Reviewer 4 Comments.

  1. Page 4 line 155, the data were...    the data was 

      Response: Thank you for your reminder. We have changed “were” in line 156 to “was”.

  1. The author should add a note that the source of the map in figure 1.

      Response:Thank you for your suggestion. We have added a note about the source of the map as follows:“the source of the map is the standard map website (Appendix A)”[lines 153-154], at the same time, the link of the website has also been added in Appendix A[Line 555].

Thank you again for your helpful comments!